**Distinguishing between early and late covering crops in the land surface model**

**Noah-MP: Impact on simulated surface energy fluxes and temperature**

Kristina Bohm[ab], Joachim Ingwersen[b], Josipa Milovac[c], Thilo Streck[b]

[a] previously published under the name Kristina Imukova

[b] Institute of Soil Science and Land Evaluation, Department of Biogeophysics, University of Hohenheim, 70593 Stuttgart, Germany

[c] Institute of Physics and Meteorology, University of Hohenheim, 70593 Stuttgart, Germany

Corresponding author: Kristina Bohm

E-Mail: imukovaks@gmail.com

**Abstract**
Land surface models are essential parts of climate and weather models. The widely used Noah-
MP land surface model requires information on the leaf area index (LAI) and green vegetation
fraction (GVF) as key inputs of its evapotranspiration scheme. The model aggregates all
agricultural areas into a land use class termed "Cropland and Pasture". In a previous study we
showed that, on a regional scale, GVF has a bimodal distribution formed by two crop groups
differing in phenology and growth dynamics: early covering crops (ECC, ex.: winter wheat,
winter rapeseed, winter barley) and late covering crops (LCC, ex.: corn, silage maize, sugar
beet). That result can be generalized for Central Europe. The present study quantifies the effect
of splitting the land use class "Cropland and Pasture" of Noah-MP into ECC and LCC on surface
energy fluxes and temperature. We further studied the influence of increasing the LCC share,
which in the study area (the Kraichgau region, southwest Germany) is mainly the result of
heavily subsidized biomass production, on energy partitioning at the land surface. We used the
GVF dynamics derived from high-resolution (5 m x 5 m) RapidEye satellite data and measured
LAI data for the simulations. Our results confirm that GVF and LAI strongly influence the
partitioning of surface energy fluxes, resulting in pronounced differences between ECC and LCC
simulations. Splitting up the generic crop into ECC and LCC had the strongest effect on land
surface exchange processes in July-August. During this period, ECC are at the senescence
growth stage or already harvested, while LCC have a well-developed, ground-covering canopy.
The generic crop resulted in humid bias, i.e. an increase of evapotranspiration by +0.5 mm d$^{-1}$
(LE: 1.3 MJ m$^{-2}$d$^{-1}$), decrease of sensible heat flux (H) by 1.2 MJ m$^{-2}$ d$^{-1}$ and decrease of surface
temperature by –1°C. The bias increased as the shares of ECC and LCC became similar. The
observed differences will impact the simulations of processes in the planetary boundary layer.
Increasing the LCC share from 28 to 38% in the Kraichgau region led to a decrease of latent heat
flux (LE) and a heating up of the land surface in the early growing season. Over the second part
of the season, LE increased and the land surface cooled down by up to 1 °C.

## 1 Introduction

Within weather and climate models, land surface exchange processes are simulated by so-called land surface models (LSMs). The main role of an LSM is to partition net radiation at the land surface into sensible heat (H), latent heat (LE) and ground heat (G) fluxes and to determine the land surface temperature. Surface energy partitioning has a significant influence on the evolution of the Atmospheric Boundary Layer (ABL). ABL evolution strongly influences the initiation of convection, cloud formation, and ultimately the location and strength of precipitation (Crawford et al. 2001, Koster et al. 2006, Santanello Jr. et al. 2013, van Heerwaarden et al. 2009, Milovac et al. 2016).

The surface energy partitioning depends on the physical and physiological properties of the land surface (Raddatz 2007). In LSMs, the earth's surface is subdivided into different land use classes, among them cropland. Physiological state variables of crops such as green vegetation fraction (GVF) and leaf area index (LAI) vary significantly throughout the growing season. This alters the biophysical parameters surface albedo, bulk canopy conductance, and roughness length, leading to significant changes in surface energy fluxes (Crawford et al. 2001, Ghilain et al. 2012, Tsvetsinskaya et al. 2001a, Wizemann et al. 2014). In many parts of the world, cropland covers a considerable part of the simulation ~~domain~~ area. Therefore, accurately simulating the seasonal variability of surface energy fluxes highly depends on an adequate representation of plant growth dynamics.

One of the widely used LSMs is Noah-MP. It is usually coupled with the Weather Research and Forecasting (WRF) model, which is intended for use from the large eddy simulation (LES) scale up to the global scale. Within each grid cell, Noah-MP computes net longwave radiation as well as LE, H and G separately for the bare soil and the vegetated tile, whereas short-wave radiation is computed over the entire grid cell (semi-tile approach; Lhomme and Chehbouni 1999, Niu et al. 2011).

Noah-MP collects agricultural areas into only general land use classes such as "Dryland Cropland and Pasture", "Irrigated Cropland and Pasture" or "Mixed Dryland/Irrigated Cropland and Pasture" etc.. Vegetation dynamics and its seasonal development are described in the Noah-MP

model by the plant variables GVF and LAI. The surface energy fluxes critically depend on accurately representing GVF and LAI dynamics (Chen and Xie 2011, Crawford et al. 2001, Refslund et al. 2014). In Noah-MP, GVF and LAI are fixed quantities: they do not depend on the weather conditions during a simulation. GVF is defined as the grid-cell fraction covered by a green canopy (Gutman and Ignatov 1998). It is a function of the upper canopy (Rundquist 2002) and represents the horizontal density of vegetation in each grid cell (Gutman and Ignatov 1998). LAI represents the vertical density of the canopy. Certain biophysical parameters in Noah-MP such as surface albedo, roughness and emissivity are considered linear functions of LAI.

By default, Noah-MP derives GVF values from the normalized difference vegetation index (NDVI) obtained from the NESDIS/NOAA satellite. These data have a resolution of 15 km × 15 km. Due to the mixing of croplands, forest and urban areas, the overall GVF is often positively biased. Moreover, as shown by Imukova et al. (2015), seasonal GVF data are strongly smoothed compared to the actual GVF dynamics. Milovac et al. (2016) and Nielsen et al. (2013) found that the GVF grid data used in Noah-MP LSM are outdated and stated that these should be updated given their importance for ABL evolution.

In a previous study, we derived GVF data with a resolution of 5 m x 5 m (Imukova et al. 2015) for a region in southwest Germany (Kraichgau) using RapidEye satellite data. On the regional scale, GVF shows a bimodal distribution mirroring the different phenology of crops. Crops could be grouped into two classes. Early covering crops (ECC), such as winter wheat, winter rape, winter barley and spring barley, develop early in spring, achieve maximum GVF usually between late May and mid-June, and become senescent in July. Late covering crops (LCC), such as corn, silage maize, and sugar beet, are drilled in spring and develop maximum ground-covering canopy from July to August. They are still green in September, when the ECC are already harvested. The dynamics of ECC and LCC vary to some degree from season to season and from region to region.

The shares of ECC and LCC may change over time, often reflecting economic decisions that may depend on policy interventions. In Germany, a substantial change in these shares was introduced by subsidizing biogas production. In 2005, 1.7 million ha of maize were cultivated in Germany.

Only 70,000 ha of this area were cropped with silage maize for biogas production (SRU Special
Report 2007). In 2009, the area cropped with maize for biogas production had increased to about
500,000 ha, while the total maize area remained almost constant (Huyghe et al. 2014). In 2012,
the total acreage of maize had increased to 2.57 million ha with 0.9 million ha intended for biogas
plants. The increase occurred mainly at the expense of grassland. Since then, the total maize crop
area has remained almost constant: 2.6 million ha in 2018 (Fachagentur Nachwachsende Rohstoffe
e. V. 2019). From 2005 to 2018, the maize area in Germany increased by about 53%.

The objectives of the present study were 1) to elucidate the extent to which surface energy fluxes
simulated with Noah-MP are affected by aggregating early and late covering crops into one generic
cropland class, and 2) to quantify the effect of a land use change, driven by the expansion of maize
cropping as a response to the increasing demand for biogas plants, on energy partitioning and
surface temperature in the Kraichgau region (southwest Germany). Additionally, we tested the
performance of the Noah-MP on LE data measured with the Eddy Covariance technique.

## 2       Materials and methods

### 2.1    *Study site and weather data measurements*

The site under study is the agricultural field belonging to the farm "Katharinentalerhof". The field
is located north of the city of Pforzheim ($48.92^0$N, $8.70^0$E). The central research site is a part of
the Kraichgau region. Kraichgau region covers about 1500 $km^2$. Mean annual temperature ranges
between 9-10° C and annual precipitation between 730 and 830 mm. The Neckar and Enz rivers
form the borders to the east. To the north and south, the region is bounded by the low mountain
ranges Odenwald and Black Forest. In the west, it adjoins the Upper Rhine Plain (Oberrheinisches
Tiefland). Kraichgau has a gently sloping landscape with elevations between 100 and 400 m above
sea level (a.s.l.). Soils predominantly formed from loess material. The region is intensively used
for agriculture: around 46 % of the total area is used for crop production. Winter wheat, winter
rapeseed, spring barley, corn, silage maize and sugar beet are the predominant crops.

Weather data used to force the Noah-MP model were acquired at an agricultural field (EC1, 14 ha) belonging to the farm "Katharinentalerhof". The terrain is flat (elevation a.s.l.: 319 m). The predominant wind direction is south-west. The study site has been described in detail in several studies (Imukova et al. 2015, Ingwersen et al. 2011, Wizemann et al. 2014).

An Eddy Covariance (EC) station was operated in the center of the EC1 field. Wind speed and wind direction were measured with a 3D sonic anemometer (CSAT3, Campbell Scientific, UK) installed at a height of 3.10 m. Downwelling longwave and downwelling shortwave radiation were measured with a NR01 4-component sensor (NR01, Hukseflux Thermal Sensors, The Netherlands). Air temperature and humidity were measured in 2 m height (HMP45C, Vaisala Inc., USA). All sensors recorded data in 30-min intervals. Rainfall was measured using a tipping bucket (resolution: 0.2 mm per tip) rain gauge (ARG100, Campbell Scientific Ltd., UK). For further details about instrumentation and data processing see Wizemann et al. (2014).

## *2.2    Eddy Covariance measurements*

In order to test the Noah-MP performance, we used the EC measurements of latent heat flux over maize (EC2) and winter wheat field (EC3) of 2012 growing season. EC2 and EC3 agricultural fields are also belonging to the farm "Katharinentalerhof" introduced above. They are 23 ha and 15 ha large. The winter wheat was planted in autumn 2011 and harvested on 29[th] of July. The maize was drilled on 2[nd] of May and harvested on 20[th] of September. The EC station was operated in the center of each field. The latent heat flux was measured in a 30-min resolution. For the maize, the LE data was only available till 20[th] of September, whereas for the winter wheat field there were no missing data. A detailed information on the EC measurements is given in Imukova et al. (2016). The EC flux data were processed with the TK3.1 software (Mauder M., 2011). Surface energy fluxes were computed from 30-min covariances. For data quality analysis we used the flag system after Foken (Mauder M., 2011). LE half-hourly values with flags from 1 to 6 (high and moderate quality data) were used to test the performance of the Noah-MP LSM. LE data was gap filled using the mean diurnal variation method with an averaging window of 14 days (Falge et al., 2001). The random error of the LE flux, which consist of the instrumental noise error of the EC station and the sampling error was computed by the TK3.1 software

(Mauder et al., 2013). For more details on EC data processing, please refer to Imukova et al.

161 (2016).


The model performance is usually tested against field measurements of sensible and latent heat
flux performed with Eddy Covariance (EC) technique (Ingwersen et al., 2011; El Maayar et al.,
2008; Falge et al., 2005). EC method is widely used method for this purpose although it has one
well-known problem. The energy balance of EC flux data is typically not closed, which means LE
and/or H fluxes measured with EC technique are most probably underestimated. Previous study
showed the EC technique provides reliable LE measurements at our study site and these data can
be used for model testing (Imukova et al. 2016).

**2.3     The Noah-MP v1.1 land surface model**
*2.3.1    Model parameterization*
Multi-physics options of Noah-MP were set as shown in the Table 1. For the simulation we used
the USGS land use dataset. The vegetation type index was set to 2 (Dryland cropland and
Pasture) and soil type index to 4 (Silt loam). The model was forced with half-hourly weather data
(wind speed, wind direction, temperature, humidity, pressure, precipitation, downwelling
longwave and shortwave radiation) measured at EC1 from 2011 to 2012. Simulations were
initialized with a spin up period of one year (2011) and run with a time step of 1800 seconds.
**Table 1.** Setting of the multi-physics options used in the Noah-MP simulation.

| Multi-physics option | Setting |
|---|---|
| Vegetation model | opt_dveg = 1: prescribed [table LAI, shdfac=FVEG] |
| Canopy stomatal resistance | opt_crs = 2: Jarvis |
| Soil moisture factor for stomatal resistance | opt_btr = 1: Noah |
| Runoff and groundwater model | opt_run = 1: SIMGM |
| Surface layer drag coefficient (CH & CM) | opt_sfc = 1: based on Monin-Obukhov similarity theory |
| Supercooled liquid water | opt_frz = 1: NY06 |
| Frozen soil permeability | opt_inf = 1: NY06 |
| Radiation transfer | opt_rad = 3: gap=1—Fveg |
| snow surface albedo | opt_alb = 2: CLASS |
| rainfall & snowfall | opt_snf = 1: Jordan91 |
| lower boundary of soil temperature | opt_tbot = 2: Noah |
| snow/soil temperature time scheme | opt_stc = 1: Semi-implicit |



*2.3.2 GVF dynamics*
The GVF data required by the Noah-MP model were derived from high-resolution (5 m x 5 m)
RapidEye satellite data. A detailed information on the deriving of the GVF data used in the current
research can be found in Imukova et al. (2015). The GVF data were calculated from the
Normalized Difference Vegetation Index (NDVI) computed from the red and near-infrared bands
of the satellite images. The relationship between GVF and NDVI was established by linear
regression using ground truth measurements. GVF maps for the Kraichgau region were derived in
a monthly resolution.
**Table 2.** GVF dynamics of early covering crops (ECC) and late covering crops (LCC) in 2012 and 2013
in the Kraichgau region, southwest Germany as well as the GVF dynamics of the generic crop.

| GVF | | 15 Apr | 15 May | 15 Jun | 15 Jul | 15 Aug | 15 Sep |
|---|---|---|---|---|---|---|---|
| GVF 2012 | ECC | - [b] | 0.74 | 0.83 | 0.37 | 0.01 [c] | 0.01 |
| | LCC | - [b] | 0.01 | 0.35 | 0.74 | 0.69 [c] | 0.56 |
| GVF 2013 | ECC | 0.54 | 0.80 | 0.57 [c] | 0.29 | 0.01 | 0.01 |
| | LCC | 0.01 | 0.06 | 0.37 [c] | 0.69 | 0.74 | 0.75 |
| **Mean GVF** | **ECC** | **0.54** | **0.77** | **0.70** | **0.33** | **0.01** | **0.01** |
| | **LCC** | **0.01** | **0.04** | **0.36** | **0.72** | **0.72** | **0.66** |
| **Generic crop GVF[a]** | | **0.39** | **0.57** | **0.60** | **0.44** | **0.21** | **0.19** |

[a] Weighted mean GVF calculated based on fractions of ECC (72%) and LCC (28%) in Kraichgau
[b] No RapidEye scenes were available for April
[c] No RapidEye scenes were available for these months, GVF values were derived by linear interpolation between adjacent months


Table 2 shows the observed and mean GVF dynamics of ECC and LCC over the growing seasons
2012 and 2013 as well as the GVF dynamics of the generic crop in the Kraichgau region. The GVF
values on the 15[th] day of each month, as required by Noah-MP model, were calculated by linearly
interpolating the monthly values derived from the GVF maps. A generic GVF dynamics was
calculated as the weighted mean of ECC and LCC from 2012 and 2013. The areal distribution of
ECC and LCC was determined from the GVF maps of May. All pixels with a GVF value below
0.5 were counted as LCC, whereas pixels with values above that threshold were assigned to ECC.
Figure 1 shows the spatial distribution of early and late covering crops in Kraichgau. The estimated
areal distribution of ECC and LCC was 72% and 28%, respectively. These results correspond well
with data of the Statistisches Landesamt Baden-Württemberg (http://www.statistik.baden-
wuerttemberg.de/ ).


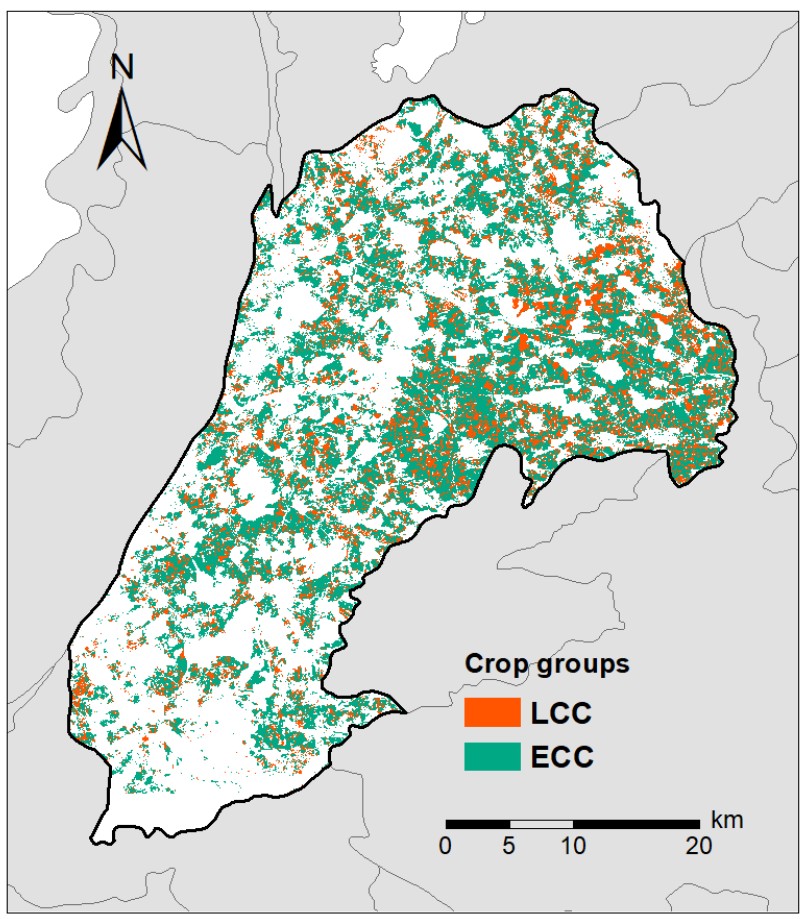

Figure 1. Map of early covering (ECC) and late covering crops (LCC) in Kraichgau region, Southwestern Germany.

*2.3.3   LAI dynamics*

Noah-MP requires prescribed LAI data for each month. Data were derived from field measurements. LAI was measured biweekly using a LAI-2000 Plant Canopy Analyzer (LI-COR Biosciences Inc., USA). In 2012 and 2013, LAI of the crops was measured on five permanently marked plots of 1 m$^2$ on three different fields. Detailed information about the study plots can be found in Imukova et al. (2015). In 2009-2011, LAI and the phenological development of the crops were measured on five permanently marked plots of 4 m$^2$ in the same three fields. The growth stages of crops were determined using the BBCH scale (Meier et al. 2009). More details on the measurements can be found in Ingwersen et al. (2011) and Ingwersen et al. (2015). Table 3 shows measured and mean LAI dynamics as well as generic LAI dynamics estimated considering shares of ECC (72%) and LCC (28%) in the study region. LAI dynamics of winter wheat and winter rape were assigned to ECC, those of maize to LCC. Mean LAI dynamic of ECC was estimated based on the measurements conducted in winter wheat and winter rape stands during the 2012 and 2013 growing seasons. Since LAI data were not available for maize in 2013, the mean LAI dynamic of LCC were assessed using field data from the same fields collected in 2009-2012.

**Table 3.** LAI dynamics of early covering crops (ECC) and late covering crops (LCC) in 2012 and 2013 in the Kraichgau region, southwest Germany, as well as the LAI dynamics of the generic crop.

| Green LAI | | 15 Apr | 15 May | 15 Jun | 15 Jul | 15 Aug | 15 Sep |
|---|---|---|---|---|---|---|---|
| LAI 2012 | ECC | 2.4 | 4.4 | 4.6 | 0.0 | 0.0 | 0.0 |
| | LCC | 0.0 | 0.1 | 0.9 | 3.2 | 5.0 | 3.7 |
| LAI 2013 | ECC | 1.7 | 4.2 | 4.3 | 0.0 | 0.0 | 0.0 |
| | LCC [b] | - | - | - | - | - | - |
| **Mean LAI** | **ECC** | **2.1** | **4.3** | **4.5** | **0.0** | **0.0** | **0.0** |
| | **LCC [c]** | **0.0** | **0.1** | **0.9** | **3.1** | **4.5** | **3.8** |
| **Generic crop LAI [a]** | | **1.5** | **3.1** | **3.5** | **0.9** | **1.3** | **1.1** |

[a] Weighted mean LAI calculated based on fractions of ECC (72%) and LCC (28%) in Kraichgau
[b] LAI data for maize in 2013 were not measured
[c] Since LAI data for maize in 2013 were not available, LAI dynamics were derived from the field data of 2009-2012 for maize in the Kraichgau region

*2.4   Simulation runs*

We firstly quantified the extent to which ECC and LCC differ with regard to their energy and water
fluxes, surface (TS) and soil temperature (TG). For this, we performed one local simulation for
each crop group using the mean LAI and the mean GVF dynamics observed during the two
growing seasons (see Table 2 and Table 3).

Secondly, to determine the effect of splitting up the vegetation dynamics of a generic crop into
that of ECC and LCC, we compared the following two local simulation runs:
Run 1: Noah-MP was forced with the GVF and LAI dynamics of the generic crop (Table 2 and
Table 3). Accordingly, in this simulation, we first computed the weighted mean of the vegetation
properties (GVF and LAI), and subsequently simulated the surface energy fluxes, TS and TG.
Run 2: We first simulated the energy and water fluxes separately for ECC and LCC with their
crop-specific vegetation dynamics. Afterwards, we calculated the weighted averages of the
simulated fluxes and temperatures based on the share of early covering (72%) and late covering
crops (28%) in Kraichgau.

Thirdly, we studied the effect of increasing the LCC share on the surface energy fluxes, surface
and soil temperatures. As mentioned in the Introduction, the maize cropping area in Germany
increased by 53% over the last decade. In our study region, this increase corresponds to a rise of
the LCC share from 28% to 38%. To study the effect of this land use change on the Noah-MP
simulations, we performed one additional generic crop simulation, but this time the generic crop
dynamics was computed with a LCC share of 38%.

*2.5    Statistical analysis*
The model performance was evaluated based on the model efficiency (EF), root mean square error
(RMSE) and bias. EF is defined as the proportion of the total variance explained by a model:

$$EF = 1 - \frac{\sum_{i=1}^{N}(P_i - O_i)^2}{\sum_{i=1}^{N}(O_i - \overline{O})^2} ,$$    Eq. 1

where $P_i$ denotes predicted values, $O_i$ and $\overline{O}$ – observed values and their mean, respectively, while
N is the number of observations. RMSE and bias were calculated as

$$RMSE = \sqrt{\frac{1}{N}\sum_{i=1}^{N}(P_i - O_i)^2}$$

Eq. 2

and

$$bias = \frac{1}{N}\sum_{i=1}^{N}(P_i - O_i).$$

Eq. 3



**3      Results**
*3.1     ECC vs. LCC*
Over the growing season, ECC and LCC show distinct differences with regard to energy
partitioning at the land surface (Figure 2). The observed shifts were strongest for LE and H. Early
covering crops already reached their maximum LE flux in May, after which LE declined during
the growing season. In contrast, LCC showed a continued increase in LE over the season, peaking
three months later in August. The smallest difference in evapotranspiration between both crops
types was on average 0.4 mm day$^{-1}$ (LE 0.9 MJ m$^{-2}$day$^{-1}$) in June, while the largest mean deviation
of -2.3 mm day$^{-1}$ (LE -5.7 MJ m$^{-2}$day$^{-1}$) occurred in August (Table 4). With regard to the H flux,
the situation was opposite (Figure 2). In the case of ECC, H flux increased continuously over the
course of the growing season, peaking in August. In contrast, LCC already reached the H
maximum in May. Afterwards, H decreased continuously until late August. As for LE, the smallest
(-1.2 MJ m$^{-2}$day$^{-1}$) and largest (5.3 MJ m$^{-2}$day$^{-1}$) mean differences in H between ECC and LCC
were observed in June and August, respectively (Table 4). Compared with LCC, the higher latent
heat fluxes of ECC in May and June resulted in a cooler land surface, on average by -2.6°C and -
1.0°C, respectively (Table 4). From July to August the situation was reversed: because latent heat
fluxes of ECC are distinctly lower than that of LCC, the surface temperature at ECC sites was up
to 4°C warmer than at LCC sites (Figure 3).

The mean difference in daily ground heat flux between ECC and LCC during the growing season ranged between -0.2 MJ m$^{-2}$ and 0.2 MJ m$^{-2}$ (Table 4). Also for the ground heat flux, the smallest difference between both crops types was observed in June (0.05 MJ m$^{-2}$).

**Table 4.** Mean differences (ECC minus LCC) in latent (LE), sensible (H) and ground heat (G) fluxes, mean surface temperature (TS) and mean ground temperature (TG) between ECC and LCC simulations.

| Month | DOY | LE mm d$^{-1}$ | MJ m$^{-2}$ d$^{-1}$ | H MJ m$^{-2}$ d$^{-1}$ | G MJ m$^{-2}$ d$^{-1}$ | TS °C | TG °C |
|---|---|---|---|---|---|---|---|
| May | 121 – 151 | 1.3 | 3.3 | -3.1 | -0.2 | -2.6 | -2.2 |
| June | 152 – 181 | 0.4 | 0.9 | -1.2 | 0.05 | -1.0 | -0.9 |
| July | 182 – 212 | -1.5 | -3.8 | 3.3 | 0.2 | 2.1 | 1.8 |
| August | 213 – 243 | -2.3 | -5.7 | 5.3 | 0.1 | 3.2 | 2.4 |
| September | 244 – 273 | -0.7 | -1.8 | 2.1 | -0.1 | 1.9 | 1.2 |

DOY - day of a year


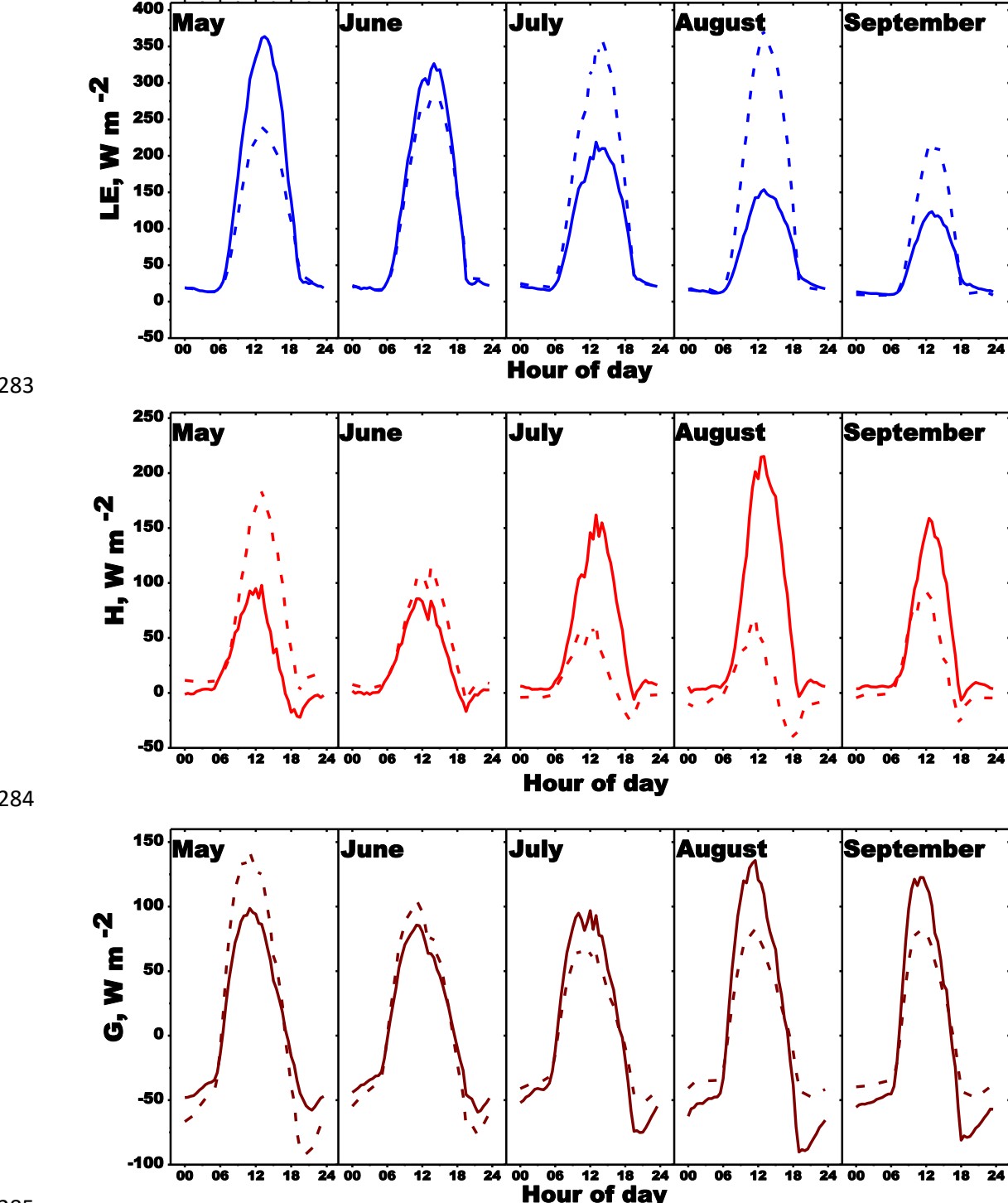



**Figure 2.** Simulation results of Noah-MP LSM for latent (LE), sensible (H) and ground heat (G) flux.
Simulations were performed for two types of crops: early covering (solid line) and late covering (dashed
line).





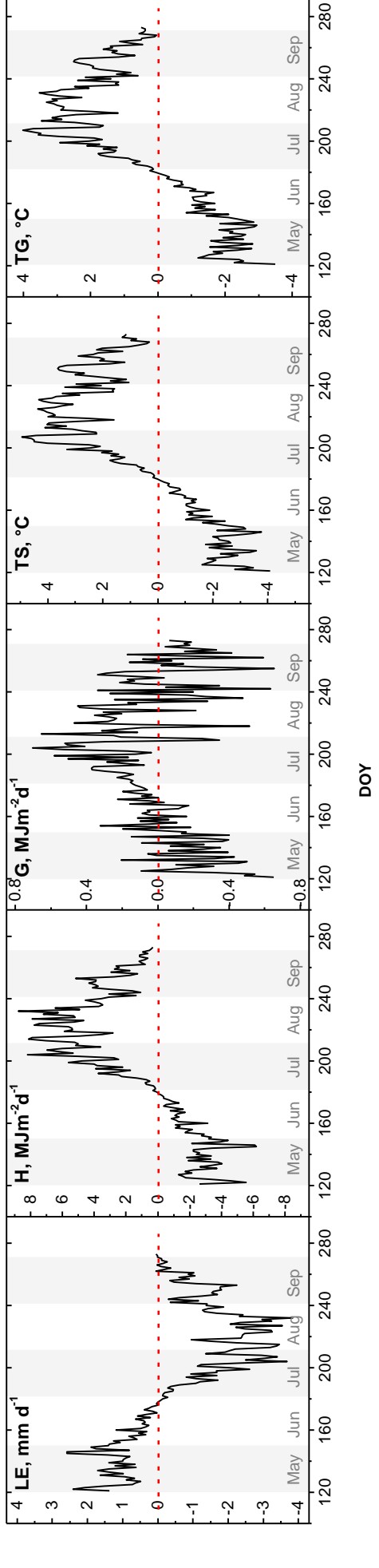

**Figure 3.** Differences (ECC minus LCC) in latent (LE), sensible (H) and ground heat (G) fluxes, mean surface temperature (TS) and mean ground temperature
(TG) between simulations for ECC and LCC.




### 3.2    Noah-MP vs. Eddy Covariance measurements

The average random error of the latent heat flux measured with EC technique for the entire growing season was about 25% over the winter wheat field and about 21% over the maize field.

The simulated latent heat flux based on ECC and LCC parametrization agreed fairly well with the Eddy Covariance data (Table 5-6, Figure 4-5). The model efficiency over the entire simulation period was 0.87 for ECC and 0.90 for LCC. The best agreement between the observations and the Noah-MP LSM using crop-type-specific sets was achieved for winter wheat in June and for maize in August and September. The generic crop parametrization showed less satisfying modeling results, particularly for the maize field (Table 5-6). For the entire growing season, EF was 0.78 for winter wheat and only 0.57 for maize. Over the winter wheat field, LE was overestimated. Overestimation of the LE was highest in in July and August. Over the maize field, LE was overestimated in May and June and underestimated in July, August and September. Particularly in May and August, the bias increased to 68.8 $Wm^{-2}$ and -56 $Wm^{-2}$, respectively. The best model performance using generic crop set was achieved for the winter in June wheat and for the maize in July.

**Table 5.** Root mean square error (RMSE), bias and modeling efficiency (EF) of the latent heat flux for the simulation runs for winter wheat stand (EC3 field).

| Variant | May | June | July | August | September | Overall |
|---|---|---|---|---|---|---|
| RMSE ($Wm^{-2}$) | | | | | | |
| ECC | 45.4 | 35.4 | 33.0 | 26.3 | 13.5 | 32.5 |
| Generic crop | 36.3 | 33.0 | 59.6 | 63.6 | 20.9 | 45.7 |
| Bias ($Wm^{-2}$) | | | | | | |
| ECC | 27.3 | 17.9 | 14.2 | 17.1 | 0.8 | 15.5 |
| Generic crop | 20.5 | 15.2 | 33.9 | 41.7 | 7.7 | 23.8 |
| EF (1) | | | | | | |
| ECC | 0.88 | 0.91 | 0.80 | 0.74 | 0.89 | 0.87 |
| Generic crop | 0.91 | 0.92 | 0.62 | 0.41 | 0.85 | 0.78 |

**Table 6.** Root mean square error (RMSE), bias and modeling efficiency (EF) of the latent heat flux for
the simulation runs for maize stand (EC2 field).

| Variant | May | June | July | August | September | Overall |
|---|---|---|---|---|---|---|
| RMSE ($Wm^{-2}$) | | | | | | |
| LCC | 53.1 | 37.3 | 31.8 | 28.1 | 18.9 | 35.7 |
| Generic crop | 102.0 | 50.9 | 29.8 | 85.8 | 43.7 | 68.0 |
| Bias ($Wm^{-2}$) | | | | | | |
| LCC | 37.4 | 21.5 | 13.7 | -14.9 | -2.5 | 11.0 |
| Generic crop | 68.6 | 29.9 | -10.6 | -56.0 | -22.9 | 1.8 |
| EF (1) | | | | | | |
| LCC | 0.59 | 0.87 | 0.94 | 0.96 | 0.96 | 0.90 |
| Generic crop | 0.30 | 0.80 | 0.91 | 0.12 | 0.77 | 0.57 |


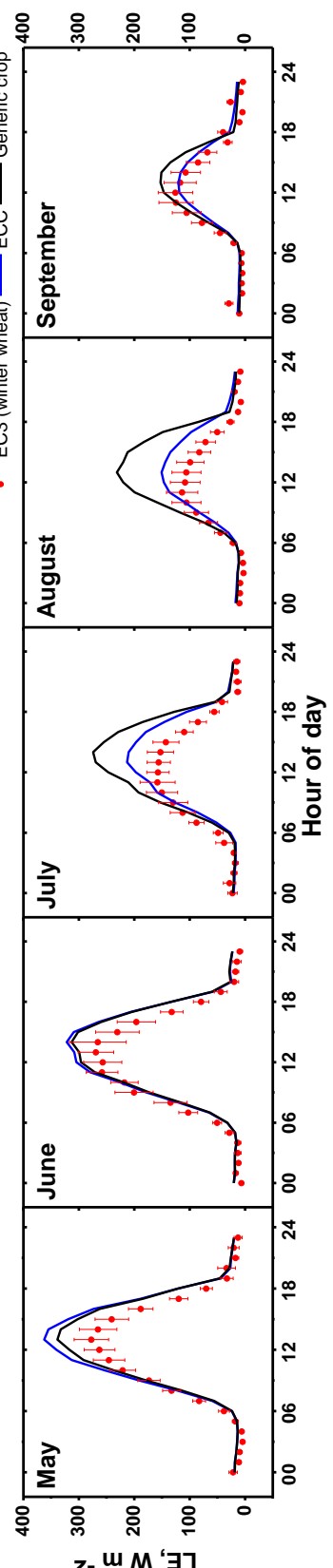


**Figure 4.** Monthly averaged measured and simulated diurnal latent heat fluxes (LE) for May to September. The Noah-MP LSM was run with two
different vegetation parametrizations: early covering crops (ECC) and generic crop.

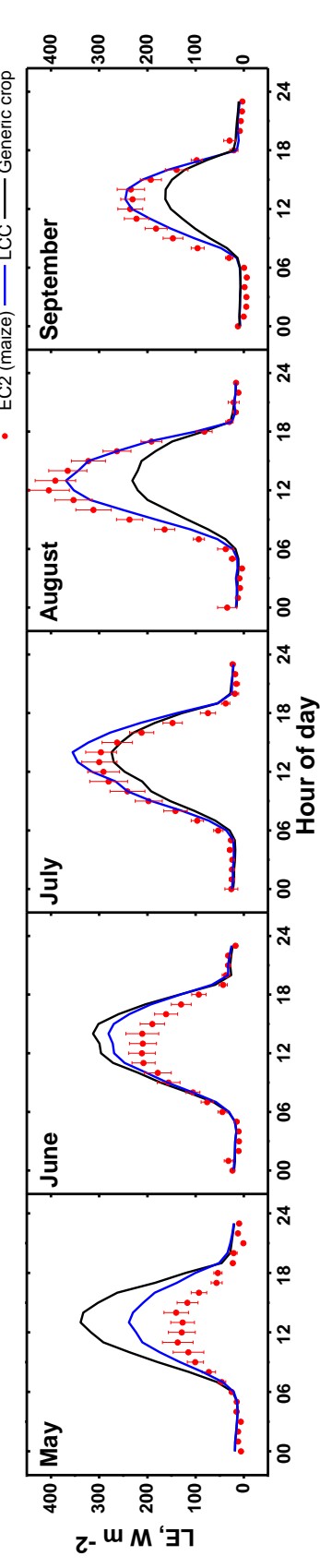


**Figure 5.** Monthly averaged measured and simulated diurnal latent heat fluxes (LE) for May to September. The Noah-MP LSM was run with two
different vegetation parametrizations: late covering crops (LCC) and generic crop.
*3.3    Run 1 vs. Run 2 (Generic crop vs. weighted mean of ECC and LCC)*
The generic crop simulation run (Run 1) generally yielded higher LE fluxes than Run 2 (i.e.
splitting up the generic crop into ECC and LCC) (Figure 6). During the growing season the mean
difference in evapotranspiration between two runs was 0.1 mm day$^{-1}$ (LE 3.7 MJ m$^{-2}$day$^{-1}$) (Table
5). Smallest mean monthly differences occurred in June and September: 0.02 mm day$^{-1}$ (LE 0.4 MJ
m$^{-2}$day$^{-1}$) and 0.03 mm day$^{-1}$ (LE 1 MJ m$^{-2}$day$^{-1}$), respectively. The most pronounced differences
in LE flux were recorded in late July (DOY 197-208) (Figure 7). The average difference in half-
hourly fluxes over this period, between 9 a.m. and 6 p.m, was 36 W m$^{-2}$, and the highest half-
hourly deviation between both runs was 83 W m$^{-2}$ (Figure 7). The highest daily deviation was 0.8
mm day$^{-1}$ (Figure 6). Over the whole season, the cumulative difference in evapotranspiration
between two runs was 20 mm, leading to a 16 percent lower seasonal water balance (SWB) in Run
1 (SWB: -133 mm) than in Run 2 (SWB: -113 mm).
**Table 5.** Mean differences in latent (LE), sensible (H) and ground heat (G) fluxes, surface
temperature (TS) and ground temperature (TG) between Run 1 and Run 2 simulations. Numbers
in brackets: the relative difference between Run 1 and Run 2 simulations in percentage.

| Month | DOY | LE mm d$^{-1}$ | MJ m$^{-2}$ d$^{-1}$ | H MJ m$^{-2}$ d$^{-1}$ | G MJ m$^{-2}$ d$^{-1}$ | TS °C | TG °C |
|---|---|---|---|---|---|---|---|
| May | 121 – 151 | 0.1 (3) | 0.3 | -0.3 (19) | -0.003 (1) | -0.3 (2) | -0.02 (0.1) |
| June | 152 – 181 | 0.02 (0.4) | 0.04 | -0.1 (4) | 0.001 (1) | -0.1 (1) | 0.01 (0.05) |
| July | 182 – 212 | 0.3 (7) | 0.6 | -0.6 (21) | -0.016 (4) | -0.4 (2) | -0.1 (0.6) |
| July* | 197 – 208 | 0.5 (14) | 1.3 | -1.2 (46) | -0.034 (10) | -1.0 (4) | -0.2 (1) |
| August | 213 – 243 | 0.2 (7) | 0.5 | -0.6 (18) | 0.004 (2) | -0.3 (1) | 0.01 (0.03) |
| September | 244 – 273 | 0.03 (1) | 0.1 | -0.2 (5) | 0.005 (3) | -0.1 (1) | 0.1 (0.4) |
| **Mean** | | 0.1 (3.7) | 0.3 | -0.4 (13.2) | -0.002 (1) | -0.2 (1.4) | -0.01 (0.1) |

DOY - day of a year



In contrast, H fluxes of Run 1 were mostly lower over all months than those simulated in Run 2
(Figure 6). From May to September, the mean difference in H fluxes was about -0.4 MJ m$^{-2}$ (-
13 %) (Table 5). The smallest difference occurred again in June, the largest difference again in
late July (Figure 7). During DOY 197-208 the mean differences in half hourly H fluxes was about
-29 W m$^{-2}$, the peak deviation being -72 W m$^{-2}$ (9 a.m.-6 p.m) (Figure 7). Cumulating these
differences over the day reduced the production of sensible heat on average in the order of
1.2 MJ m$^{-2}$, corresponding to a 46 % reduction compared to Run 2 (Table 5). Ground heat fluxes
as well as soil temperature were affected only moderately by the different vegetation
parameterization of Run 1 and 2 (Figure 7, Figure 6). As for LE and H, the largest mean differences
in G fluxes were observed during DOY 197-208 ( -0.034 MJ m$^{-2}$ = 10%) (Table 5).

Due to the humid bias of Run 1, the canopy surface was cooler than in Run 2 in all months. On
average, TS of Run 1 was 0.2 °C (~1.4%) lower during the growing season than in Run 2. In late
July (DOY 197-208) the mean daily difference was -1 °C (Table 5, Figure 6) and reached a daytime
(9a.m.-6p.m.) peak difference of up to -2.6 °C (Figure 7).

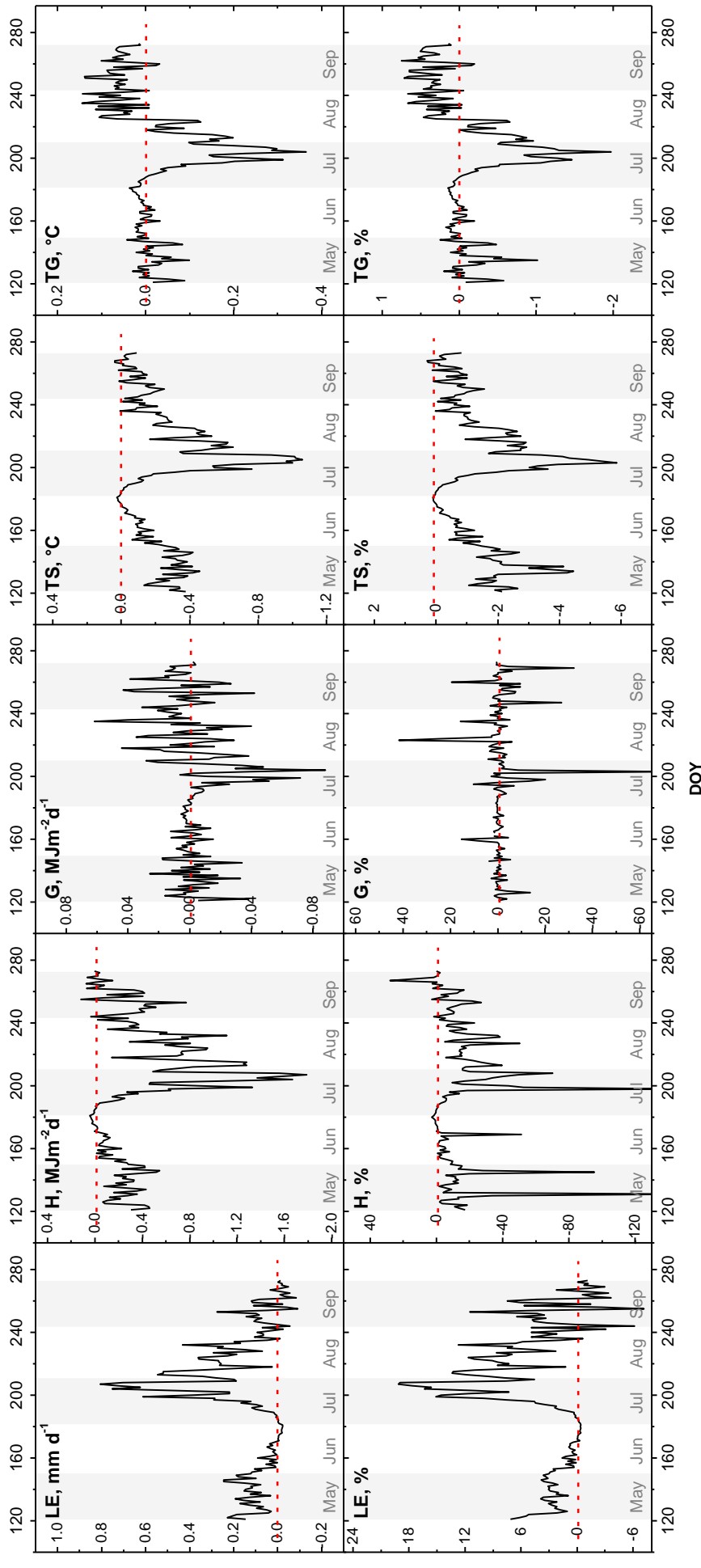

**Figure 6.** Differences in latent (LE), sensible (H) and ground heat (G) fluxes, mean surface temperature (TS) and mean ground temperature (TG) between Run 1 and Run 2 simulations (*Run 1 - Run 2*). Given percentages are relative differences between Run 1 and Run 2 simulations.


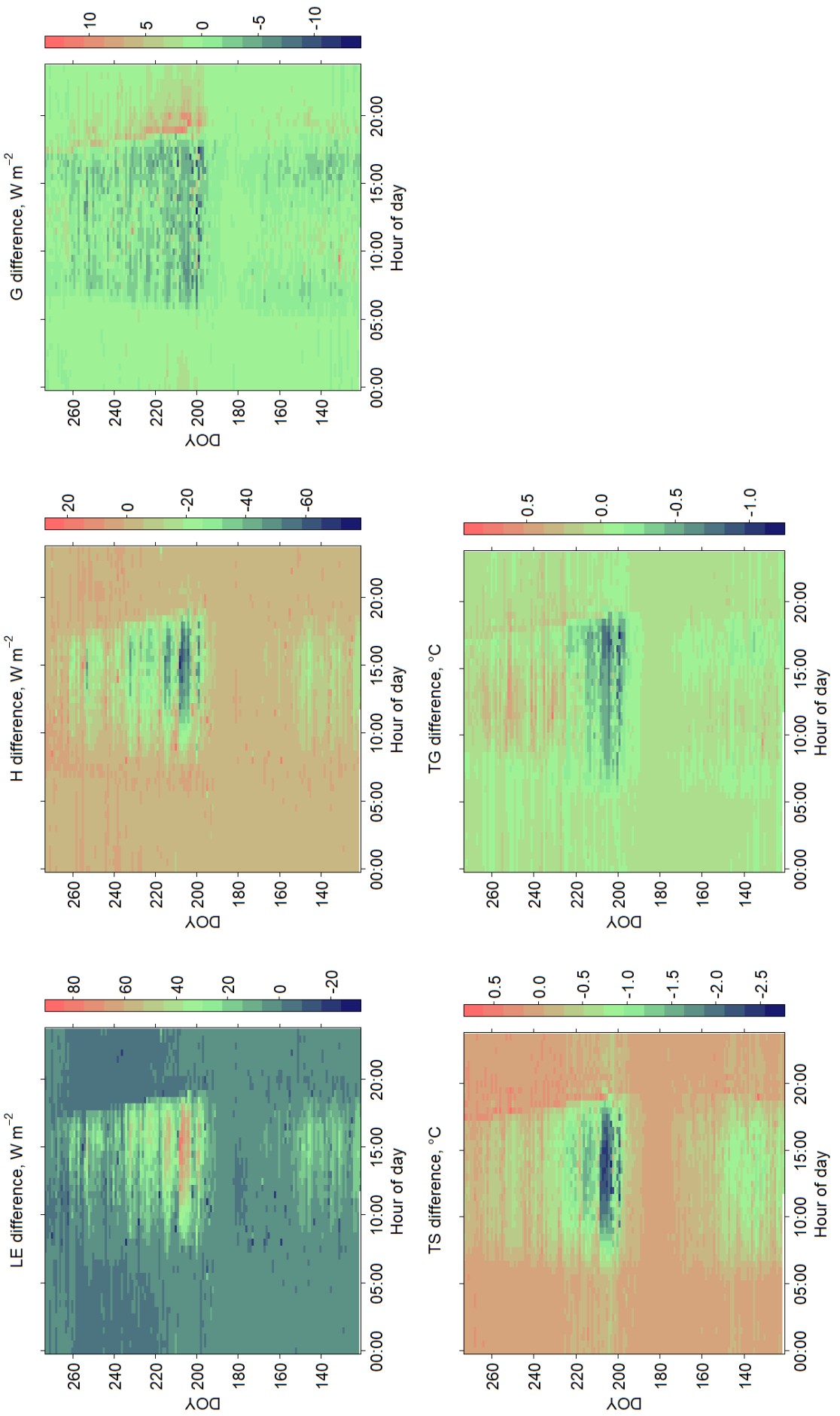

**Figure 7.** Differences in latent (LE), sensible (H) and ground heat (G) fluxes, mean surface temperature (TS) and mean ground temperature (TG) between Run 1 (generic crop) and Run 2 (weighted mean of early and late covering crops) simulations *(Run 1 - Run 2)*.

### 3.4    Land use change towards LCC

Increasing the LCC fraction from 28% to 38% mainly led to changes in LE and H fluxes (Table 6). That LCC increase lowered the LE flux (-0.3 MJ m$^{-2}$ day$^{-1}$-or ET 0.1 mm day$^{-1}$) early in the season. This was accompanied by a higher H flux (+0.3 MJ m$^{-2}$ day$^{-1}$), which in turn led to a 0.3 °C warmer surface temperature than for the runs with the actual ECC-LCC ratio. From July to September, increasing the LCC fraction boosted evapotranspiration by about 0.2 mm day$^{-1}$ (LE 0.4 MJ m$^{-2}$ day$^{-1}$) and decreased the H flux by about 0.3 MJ m$^{-2}$ day$^{-1}$ (Table 6). The largest half-hourly differences occurred in August (DOY 213-243, Figure 8), amounting to +40 W m$^{-2}$ and -30 W m$^{-2}$ for LE and H, respectively. The smallest deviations for both fluxes were recorded in June. Over the July–September period, the higher LE flux of the simulation run with the increased LCC fraction cooled the land surface up to -1 °C (Figure 8). In general over the growing season, increasing the LCC share by 10% led to an increase in cumulative evapotranspiration, which in turn resulted in a 10 mm lower seasonal water balance (SWB: -143 mm).

With regard to the ground heat flux, increasing the LCC fraction led to an up to 10 W m$^{-2}$ higher flux over the noon time during the second part of the growing season (Figure 8), whereas early in the season the differences did not exceed 0.2°C (Table 6).

**Table 6.** Mean differences in latent (LE), sensible (H) and ground heat (G) fluxes, surface temperature (TS) and ground temperature (TG) between simulations with the LCC fraction increased by 10 % and the baseline simulation *(increased LCC share minus baseline simulation)*. Numbers in brackets: the relative difference between *increased LCC share and baseline simulation* in percentage

| Month | DOY | LE mm d$^{-1}$ | MJ m$^{-2}$d$^{-1}$ | H MJ m$^{-2}$d$^{-1}$ | G MJ m$^{-2}$d$^{-1}$ | TS °C | TG °C |
|---|---|---|---|---|---|---|---|
| May | 121 – 151 | -0.1 (3.3) | -0.3 | 0.3 (14) | 0.02 (1) | 0.3 (2) | 0.2 (1) |
| June | 152 – 181 | -0.04 (1.0) | -0.1 | 0.1 (6) | -0.005 (0.5) | 0.1 (1) | 0.1 (1) |
| July | 182 – 212 | 0.2 (4.3) | 0.4 | -0.3 (12) | -0.02 (6) | -0.2 (1) | -0.2 (1) |
| August | 213 – 243 | 0.2 (7.6) | 0.6 | -0.5 (17) | -0.01 (1) | -0.3 (2) | -0.2 (1) |
| September | 244 – 273 | 0.1 (3.8) | 0.2 | -0.2 (4) | 0.01 (4) | -0.2 (1) | -0.1 (1) |

DOY - day of a year

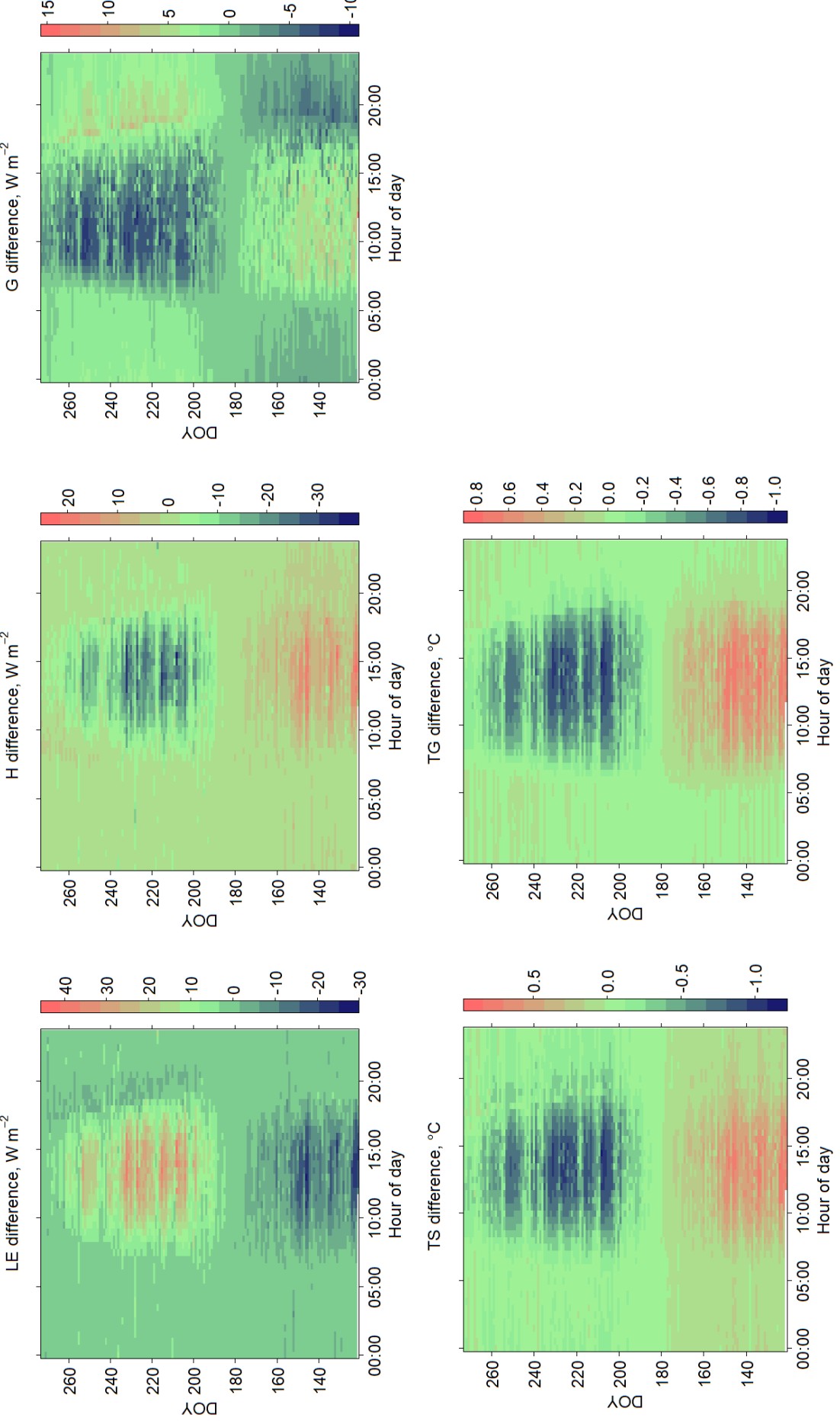

**Figure 8.** Impact of increasing the LCC fraction from 28% to 38% on latent (LE), sensible (H) and ground heat (G) fluxes, surface temperature (TS) and ground temperature (TG) (*Increased LCC share minus baseline simulation*).

## 4 Discussion

The comparison of the ECC and LCC simulations confirmed that GVF and LAI significantly affect the partitioning of surface energy fluxes. LE flux increases with crop growth and peaks when the canopy is fully developed, i.e. have maximum LAI and GVF. By contrast, the highest H and G fluxes were observed at sparsely covered fields or on the fields with a senescent canopy. During the main growth period of crops, H and G fluxes were quite low. ECC and LCC crops vary significantly in sowing and harvest date, leaf area and senescence dynamics, water use efficiency and phenology. Their surface energy fluxes therefore differ distinctly. Our simulation results are in agreement with experimental data of Wizeman et al. (2014) as well as with modeling studies of Sulis et al. (2015), Tsvetsinskaya et al. (2001b), Xue et al. (1996) or Ingwersen et al. (2018).

Simulation results based on ECC and LCC parametrization are in complete harmony with the field observations at our study site. The performance test of the Noah-MP on the EC data showed the crop-type-specific sets significantly improve the simulation of latent heat flux at the field scale. In contrast, generic crop parametrization showed less satisfying modeling results. In general, it performed better for winter wheat stand than for maize. Based on the generic crop set, simulation results tend to greatly overestimate the latent heat flux for maize in the beginning of the growing season when the plants are small. In August and September, the latent heat flux was in contrast distinctly underestimated, during this period the maize canopy is fully developed. For wheat, model overestimates the latent heat flux, particularly during July – September period, when the winter wheat stand ripened and senescence or harvested.

Besides the vegetation dynamics, the simulated energy and water fluxes depend on additional model settings as well. Ingwersen et al. (2011) performed a sensitivity study with the Noah model for our study site. He found that among the vegetation parameters the minimum stomatal resistance (RS) and a parameter used in the radiation stress function of the Jarvis scheme (RGL) are the most sensitive parameters. Using constant RS, as it is implemented in Noah, results in the underestimation of sensible heat flux and overestimation of latent heat flux during the ripening stage of the cereals. Considering a monthly varying RS helped to distinctly improve the simulation of the energy and water fluxes at the land surface. Ingwersen et al. (2010) concluded, integrating the crop growth model which delivers daily RS, LAI and GVF values into Noah would greatly

enhance the overall performance of the land surface model. Among the soil parameters, the most sensitive parameters are the soil moisture threshold where transpiration begins to stress (REFSMC), maximum soil moisture content (MAXSMC) and soil moisture threshold where direct evaporation from the top layer ends (DRYSMC). Considering these parameters has also a potential to further improve of simulation results.

The potential increase of the LCC fraction (driven by the high demand for biogas and forage production) leads to significant changes in the partitioning of the energy fluxes at the croplands. In recent years the total area under maize in Germany has more than doubled. This corresponds to an approximately 10% increase of the LCC fraction for the study region. In the early vegetation period, the altered ECC-LCC ratio leads to a decrease of evapotranspiration, an increase of H fluxes, and a warmer cropland surface because, during that period, a higher fraction of fields is bare or sparsely covered with vegetation. In mid-June, the situation reverses. The higher share of LCC boosts LE fluxes, decreases H fluxes and lowers surface temperatures. The increased evapotranspiration over the growing season, in turn, leads to a lower seasonal water balance.

Comparing the generic crop simulation (Run 1) with the weighted mean of two separate simulations for ECC and LCC (Run 2) showed the largest difference over the second half of the growing season, particularly during late July/early August. In July, ECC become senescent: GVF drops sharply and green LAI equals zero. In early August, ECC are usually harvested. In contrast, LCC have a developed ground-covering canopy during July-August. Leaves of these crops are still green in September. This transition period is very smooth in the case of the generic crop, resulting on average in about 14 % higher LE and in about 46%, 10% and 4% lower H, G and surface temperature, respectively, compared with Run 2.

The results presented above apply to the ECC-LCC ratio within our study area. What can we expect in agricultural landscapes with different ECC-LCC ratios? The ECC-LCC ratio has nearly no effect on energy partitioning in June, whereas in May, July and August its influence on the turbulent fluxes is pronounced (Figure 9). The weak effect in June is because, during this period, the LAI and GVF of ECC and LCC are similar (Figure 11). In the other months, however, the ECC-LCC ratio heavily affects the energy partitioning. For example, increasing the LCC share from 10% to

90% boosts daily evapotranspiration in August from 2.5 mm d$^{-1}$ to 4.3 mm d$^{-1}$, decreases the H
flux by about 4.1 MJ m$^{-2}$ d$^{-1}$ and cools down the cropland surface by 2 °C. Over the growing season,
the increase in the LCC share leads to a general increase in evapotranspiration, which in turn
lowers the seasonal water balance (Table 7). Moreover, different ECC-LCC ratios will also affect
the above-mentioned humid bias of the generic crop parameterization (Figure 10). The bias is
largest if ECC and LCC shares are balanced (ECC 50% and LCC 50 %), whereas combinations
with one predominant crop distinctly lower the bias. In August, for instance, the LE differences
between the two runs with ECC 50% - LCC 50% equal 0.27 mm day$^{-1}$, while ECC 10% - LCC 90
% yields differences of 0.09 mm day$^{-1}$.
**Table 7.** Weather data and simulation results of Noah-MP LSM for cumulative evapotranspiration for the
Kraichgau region. Simulations were performed considering different shares of early covering crops
(ECC) and late covering crops (LCC).

| ECC and LCC shares | Total rainfall (R), mm | Cumulative evapotranspiration (ET), mm | Seasonal water balance (R-ET), mm |
|---|---|---|---|
| ECC 90% LCC 10% | 388 | 496 | -108 |
| ECC 70% LCC 30% | 388 | 522 | -134 |
| ECC 50% LCC 50% | 388 | 544 | -156 |
| ECC 30% LCC 70% | 388 | 557 | -169 |
| ECC 10% LCC 90% | 388 | 563 | -175 |


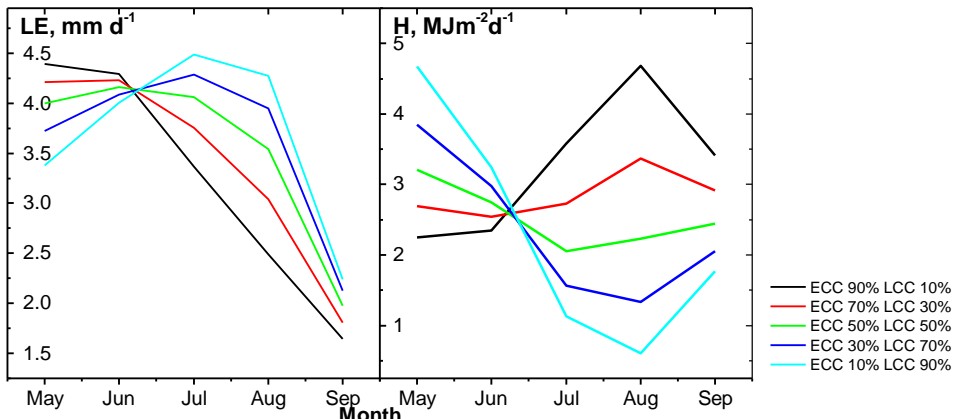


**Figure 9.** Simulation results of Noah-MP LSM for latent (LE) and sensible (H) heat flux. Simulations
were performed considering different shares of ECC and LCC.

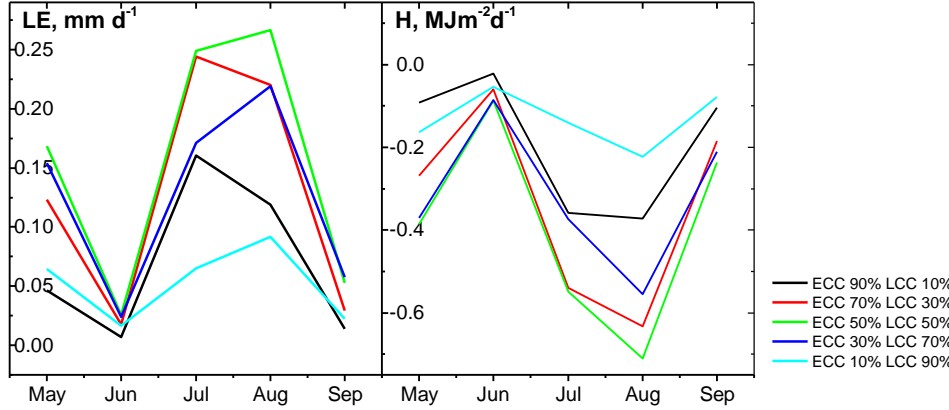


**Figure 10.** Differences in latent (LE) and sensible heat (H) fluxes between Run 1 and Run 2 simulations *(Run 1 - Run 2)*. Simulations were performed considering different shares of ECC and LCC.

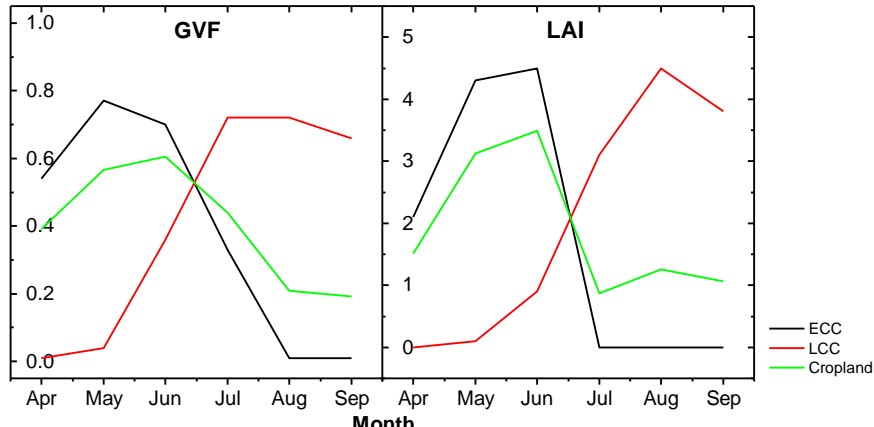


**Figure 11.** GVF and LAI dynamics of early covering crops (ECC), late covering crops (LCC) and Cropland.

Our results show that performing simulations based on single dynamics for each type of crop (ECC and LCC) improve simulations of surface fluxes during transition periods and at the end of the growing season. Lumping ECC and LCC into one land-use class (Croplands and Pasture), as done in Noah-MP, is an oversimplification. Several authors demonstrated the necessity to distinguish biophysical plant parameters between substantially different crops to obtain representative simulation results in the lower atmosphere (Sulis et al. 2015, Tsvetsinskaya et al. 2001b, Xue et al. 1996). They showed that high-resolution spatial information on various croplands and associated physiological characterizations can significantly improve the simulations of land surface energy fluxes, leading to better weather and climate predictions.

Changes of LAI and GVF with plant growth lead to changes in surface albedo, bulk canopy
conductance and roughness length, which in turn alter the partitioning of surface energy fluxes
(Chen and Xie 2011, Chen and Xie 2012, Crawford et al. 2001, Tsvetsinskaya et al. 2001a, Xue et
al. 1996). Such altered energy partitioning at the land surface then changes the thermodynamic
state of the atmospheric boundary layer withregard to air temperature, surface vapor pressure,
relative humidity and finally rainfall (Chen and Xie 2012, McPherson and Stensrud 2005, Sulis et
al. 2015, Tsvetsinskaya et al. 2001b). The observed differences between Run 1 and crop-type-
based runs will most probably influence the simulated processes in the ABL. For instance, Sulis
et al. (2015) significantly improved the simulations of land surface energy fluxes by using the
crop-specific physiological characteristics of the plant. They observed a difference of about 40%
between simulated fluxes using the generic and crop-specific parameter sets. The differences in
the land surface energy partitioning led to different heat and moisture budgets of the atmospheric
boundary layer for the generic and specific (sugar beet and winter wheat) croplands. In the case of
specific croplands, particularly sugar beet, those authors observed a larger contribution of the
entrainment zone to the heat budget of the ABL as well as a shallower ABL.

McPherson and Stensrud (2005) examined the impact of directly substituting the tallgrass prairie
land use class with winter wheat on the formation of the ABL. These crops have different growing
seasons. In the U.S. Great Plains, native prairie tallgrass mainly grows in summer, while winter
wheat grows throughout winter and reaches maturity in late spring. Simulations showed a larger
LE and lower H over the area with the winter wheat stand in comparison with tallgrass. By 2100
UTC, LE ranged from 300 to 400 W m$^{-2}$ for the wheat run and from 200 to 275 W m$^{-2}$ for the
tallgrass run. H ranged from 25 to 125 W m$^{-2}$ for the former and from 100 to 200 W m$^{-2}$ for the
latter. Substituting tallgrass prairie with winter wheat boosted the atmospheric moisture near the
surface above- and downstream of the study area, and resulted in a shallower ABL above- and
downstream of this area. The shallower ABL reduced the entrainment of higher-momentum air
into the ABL and therefore led to weaker winds within the ABL.

Milovac et al. (2016) performed six simulations at 2 km resolution with two local and two nonlocal
ABL schemes combined with two LSMs (Noah and Noah-MP) to study the influence of energy
partitioning at the land surface on the ABL evolution on a diurnal scale. They observed that LE
simulated by Noah-MP was more than 50% lower than that simulated by Noah. As expected, a
lower LE resulted in a drier ABL. The ABL evolution and its features strongly influence the
initiation of convection and cloud formation as well as the location and strength of precipitation.
For instance, drier and higher ABL would yield a higher lifting condensation level, leading to
higher clouds and a higher probability of convective precipitation.

## 5    Conclusions

GVF and LAI significantly affect the simulation of energy partitioning, yielding pronounced differences between simulated surface energy and water fluxes and temperatures of ECC and LCC. In our study area, the use of a generic crop parameterization (Croplands and Pasture in Noah-MP) resulted in a humid bias along with lower surface temperatures. This humid bias will be largest in landscapes with a balanced share of ECC and LCC, whereas in landscapes in which one of the two crop types predominate, the bias will be weaker. We observed the strongest effects on turbulent fluxes over the second part of the season, particularly in July-August. During this period, ECC are at senescence growth stage or already harvested, while LCC have a fully developed ground-covering canopy. We therefore expect that the observed differences will impact the simulation of processes in the ABL. Our results show that splitting up croplands into ECC and LCC can improve LSMs, particularly during transition periods and late in the growing season.

Increasing the LCC fraction by 10% reduces evapotranspiration and increases surface temperatures over the first part of the growing season. Later in the season, this land use change leads to the opposite situation: increased evapotranspiration accompanied by a slight cooling of the land surface. Over the growing season, an increase of the LCC share by 10% leads to higher cumulative evapotranspiration, which in turn lowers the seasonal water balance.

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
