# Peer review of "Distinguishing between early and late covering crops in the land surface model"

_Biogeosciences, 2019_

## Referee Comment (RC1) · Anonymous Referee #1 · 8 Jan 2020

Bohm et al., investigate the impacts of including early versus late cover crops in Noah-MP to model surface-atmosphere fluxes in an agricultural region in Germany. The manuscript is overall very well written and I have only minor comments. As Noah-MP is getting used more often in regional climate simulations, the authors demonstrate a good method for improving the seasonal agriculture dynamics and land use representation within Noah-MP. There are a couple areas that I think could be improved upon.

1: A map of the Kraichgau region of Germany with the accompanying GVF data and the spatial representation of ECC vs. LCC would help readers conceptualize the study

region and better understand what a 10% increase in LCC share means.

2. The weather data driving Noah-MP is derived from the study site EC1. Comparing the surface energy fluxes calculated by Noah-MP to observations with the eddy covariance instrumentation at EC1 would aid readers in understanding how improved (by splitting crops into ECC and LCC) the surface fluxes are compared to generic crop representations included with Noah-MP.

3. It's difficult to discern whether Noah-MP is being run only for the study site EC1 (point location) or for the entire Kraichgau region. The authors state Noah-MP simulations were performed for the entire Kraichgau region but Table 2 shows GVF dynamics for only for 1 point. If it's for a single point location, then more language is needed to clarify this. If it's for the entire region then a justification for using weather data acquired at one point location for simulating the energy fluxes of the entire Kraichgau region is needed. A discussion of the spatial resolution of Noah-MP would then be needed as well.

4. Since eddy covariance data exists for the site EC1, discussion about how the other Noah-MP parameters (included in Noah-MP parameter table) might influence the results when vegetation type is set to 2 such as SAI or roughness length.

---

## Referee Comment (RC2) · Anonymous Referee #2 · 15 Jan 2020

This manuscript describes how incorporating leaf area index and green vegetation fraction for specific crop groups effects the output of land surface model. Results from Bohm et al. indicate that the partitioning land cover into early and late cover crops is important for regulation climate modeling simulations. These are results are intriguing and consistent and clearly communicated by the text and the figures of the paper. Overall, this is a nice and compelling story but I think the authors should significantly clarify section 2.3 (simulation runs) to avoid confusion. This would also strengthen the results section.

There are several things that need clarification in section 2.3

1. How long were the Noah-MP simulations run: for a single year or from 2012-2013? Are the results plotted in the figures results from Noah-MP 2012 and 2013 output or results from Noah-MP driven with the multi growing season mean?

2. In the second set of simulations there are two runs, the "generic crop" and run 2. Is the second run "crop specific", the weighted average of the Noah-MP driven separately with just LCC and just ECC LAI and GVF dynamics?

3. In the runs used as results in section 3.3 is the LCC share increasing over time or was this additional run driven with a "generic crop" equivalent that used a different share weight?

Specific Comments

1. Line 36: The acronym LE is used without being defined first

2. Line 57: What do you mean by simulation domain? Is that the Kraichgau region?

3. Line 202: What was the weighted average weighted by? The crop type area?

4. Line 425: The idea in this sentence seems incomplete, the GVF and LAI yields pronounced differences between what, the crop types or atmospheric flux from the crop types?

---

## Author Comment (AC1) · 8 Mar 2020

We thank the reviewer for his very positive feedback and helpful comments. We have addressed all the questions and comments as described below.

1: A map of the Kraichgau region of Germany with the accompanying GVF data and the spatial representation of ECC vs. LCC would help readers conceptualize the study region and better understand what a 10% increase in LCC share means.

We added a figure with a spatial representation of ECC and LCC crops in the Kraichgau

region. Please see Figure 1 (Line 206)

2. The weather data driving Noah-MP is derived from the study site EC1. Comparing the surface energy fluxes calculated by Noah-MP to observations with the eddy co-variance instrumentation at EC1 would aid readers in understanding how improved (by splitting crops into ECC and LCC) the surface fluxes are compared to generic crop representations included with Noah-MP.

We tested the Noah-MP performance against latent heat flux measured with the Eddy Covariance method and added the obtained results into the Manuscript. Please see Chapters 2.2, 2.5, 3.2 and lines 401-410 of the discussion part.

3. It's difficult to discern whether Noah-MP is being run only for the study site EC1 (point location) or for the entire Kraichgau region. The authors state Noah-MP simula-tions were performed for the entire Kraichgau region but Table 2 shows GVF dynamics for only for 1 point. If it's for a single point location, then more language is needed to clarify this. If it's for the entire region then a justification for using weather data ac-quired at one point location for simulating the energy fluxes of the entire Kraichgau region is needed. A discussion of the spatial resolution of Noah-MP would then be needed as well.

Obviously, this methodological point was misleadingly described in the manuscript. The simulations were forced with the local weather data of EC1 and Noah-MP was informed with regionally-derived GVF and LAI data of the Kraichgau. We removed the sentence "Noah-MP simulations were performed for the Kraichgau region", which was obviously misleading and rephrased it in "The site under study is the agricultural field belonging to the farm "Katharinentalerhof". The field is located north of the city of Pforzheim (48.920N, 8.700E). The central research site is a part of the Kraichgau region." (line 119-121).

4. Since eddy covariance data exists for the site EC1, discussion about how the other Noah-MP parameters (included in Noah-MP parameter table) might influence the

results when vegetation type is set to 2 such as SAI or roughness length.

In the discussion, we added a study of Ingwersen et al. (2011) who performed a sensitivity study with the Noah model for our study site. Please see lines 412-425.
* * *

---

## Author Comment (AC3) · 8 Mar 2020

We thank the reviewer for his very positive feedback and helpful comments. We have addressed all the questions and comments as described below.

1. How long were the Noah-MP simulations run: for a single year or from 20122013? Are the results plotted in the figures results from Noah-MP 2012 and 2013 output or results from Noah-MP driven with the multi growing season mean?

The Noah-MP was run over two years 2011-2012. The first year (2011) is intended

to be a 'warm-up' period; thus, the following year, 2012, was considered for the assessment of the impact of different crop groups on simulated surface energy fluxes and temperature. The results plotted in the figures results from Noah-MP 2012 output. Unfortunately, there was a typo on line 177, 2013 was written instead of 2012. We corrected it now.

2. In the second set of simulations, there are two runs, the "generic crop" and run 2. Is the second run "crop specifįc", the weighted average of the Noah-MP driven separately with just LCC and just ECC LAI and GVF dynamics?

In the second run, we performed two simulations: one for early covering crops using their specific LAI and GVF dynamics and another one for late covering crops using corresponding LAI and GVF dynamics. Afterward, we calculated the weighted average of the simulated fluxes and temperatures considering the spatial distribution of early covering (72%) and late covering crops (28%) in the study region. For greater clarity, we rewrote lines 238-241 in the manuscript. It reads now as follows: "Run 2: We first simulated the energy and water fluxes separately for ECC and LCC with their crop-specific vegetation dynamics. Afterward, we calculated the weighted averages of the simulated fluxes and temperatures based on the share of early covering (72%) and late covering crops (28%) in Kraichgau".

3. In the runs used as results in section 3.3 is the LCC share increasing over time or was this additional run driven with a "generic crop" equivalent that used a different share weight?

To study the effect of increasing the LCC share from 28% to 38% in the study region on the Noah-MP simulations, we performed one additional generic crop simulation, but this time the generic crop dynamics were computed with an LCC share of 38% (Please, see lines 246-248 of the manuscript). Similar simulations were presented also in the discussion part of the manuscript. In lines 446-460, we show the results for five additional simulations using generic crop but with different ECC-LCC shares: ECC

90%-LCC 10%, ECC 70%-LCC 30%, ECC 50%-LCC 50%, ECC 30%-LCC 70%, ECC 10%-LCC 90%).

Specific Comments 1. Line 36: The acronym LE is used without being defined first

We corrected it (Line: 37)

2. Line 57: What do you mean by simulation domain? Is that the Kraichgau region?

In this case, the simulation domain can be also read as a simulation area. On line 57, we mean that in many parts of the world, cropland as a land-use class in the land surface models can cover a considerable part of the simulation area. We used now the word 'area' on line 58 as it fits better.

3. Line 202: What was the weighted average weighted by? The crop type area?

We corrected it. Please see above and the lines 238-241 of the manuscript.

4. Line 425: The idea in this sentence seems incomplete, the GVF and LAI yields pronounced differences between what, the crop types or atmospheric flux from the crop types?

We corrected it. Now it reads as "GVF and LAI significantly affect the simulation of energy partitioning, yielding pronounced differences between simulated surface energy and water fluxes and temperatures of ECC and LCC." (Please, see lines: 525-526)